# Sedentary Behaviors and Eating Habits in Active and Inactive Individuals: A Cross-Sectional Study in a Population from Southern Italy

**DOI:** 10.3390/bs14030208

**Published:** 2024-03-05

**Authors:** Patrizia Calella, Mirella Di Dio, Concetta Paola Pelullo, Gabriella Di Giuseppe, Fabrizio Liguori, Giovanna Paduano, Giuliana Valerio, Giorgio Liguori, Francesca Gallè

**Affiliations:** 1Department of Medical, Movement, and Wellbeing Sciences, University of Naples “Parthenope”, 80133 Naples, Italy; mirella.didio@collaboratore.uniparthenope.it (M.D.D.); concettapaola.pelullo@uniparthenope.it (C.P.P.); giuliana.valerio@uniparthenope.it (G.V.); giorgio.liguori@uniparthenope.it (G.L.); francesca.galle@uniparthenope.it (F.G.); 2Department of Experimental Medicine, University of Campania, “Luigi Vanvitelli”, 80138 Naples, Italy; gabriella.digiuseppe@unicampania.it (G.D.G.); giovanna.paduano@studenti.unicampania.it (G.P.); 3Department of Economics and Legal Studies, University of Naples “Parthenope”, 80132 Naples, Italy; fabrizio.liguori@studenti.uniparthenope.it

**Keywords:** dietary habits, sedentary behaviors, physical activity, exercise, sport, gym practitioners, athletes

## Abstract

The aim of this study was to analyze sedentary behaviors and dietary habits assumed by individuals regularly practicing exercise in a gym, sports athletes and inactive individuals. The Sedentary Behavior Questionnaire and the Healthy Dietary Habits Index were administered online to evaluate the time spent in sedentary activities during the week and the habits of food consumption among adult individuals from the Campania region, in the south of Italy. Of the 411 participants, 25% were inactive, 34% were gym practitioners and 41% practiced different sport disciplines. Significant differences were found for sedentary habits adopted during the week and diets between athletes and inactive participants. However, no significant differences were observed for sedentary activities on the weekend and some sedentary behaviors such as video gaming or working/studying during the week. With regard to diet, athletes showed healthier food choices, such as fruit and vegetable consumption. The findings of this study underline the need for enhancing the awareness of the local population regarding the detrimental effects of unhealthy dietary behaviors and sedentary time, especially but not exclusively among inactive individuals.

## 1. Introduction

According to the World Health Organization, as many as 3.2 million deaths a year are attributable to a sedentary lifestyle [1]. In adults, higher amounts of sedentary behaviors (SBs) are associated with several adverse outcomes, such as all-cause mortality, cardiovascular disease and cancer mortality, and higher incidences of cardiovascular disease, cancer and type 2 diabetes [2]. The term sedentary lifestyle refers to the time devoted to activities with energy expenditure ≥1.5 Metabolic Equivalent of the Task—MET in a sitting or lying position [3]. In the adult population, most of the time of daily life is spent adopting SB, such as PC use, TV viewing, recreational activities related to screen use or driving [4], whereas physical inactivity (PI) is referred to as an insufficient physical activity (PA) level to meet present recommendations [2]. Therefore, SB is a component of PI.

There is a substantial difference between engaging in SBs and not engaging in PA. On the other hand, playing sports or doing exercise for some hours during the week does not make one exempt from leading a sedentary life and therefore from implementing typical sedentary behaviors in daily life and for most of the day. PI speaks to a failure to meet the recommended levels of overall bodily movement for health, whereas SBs specifically pertain to low-energy activities performed while sitting or lying down. Importantly, the distinction between engaging in PA and adopting an active lifestyle emphasizes the significance of incorporating movement into daily routines beyond structured exercise sessions. This nuanced understanding is crucial for designing effective health interventions and promoting well-being.

Changes in sedentary habits may not correspond with changes in moderate to vigorous leisure-time physical activity since SBs and PA are actually two separate and non-opposing functional constructs [5]. In particular, it appears that SBs are associated with adverse health outcomes different than those attributed to the lack of PA [6].

Furthermore, SBs and PAalso seem to influence eating habits in different ways; those who exercise are oriented towards healthier food choices, while those who adopt SBstend to make unhealthy food choices [7]. Typical SBs, such as spending time in front of a screen or watching TV, have been associated with low consumption of fruits and vegetables; heavy consumption of high-energy snacks, beverages and fast food; and high total energy intake [8].

With regard to this, it should be taken into consideration that the use of electronic devices consistently exposes people to advertisements for high-energy, low-nutritional-value foods [9]. Therefore, the health risks related to sedentary habits are added to those resulting from an incorrect diet, mainly related to the development of obesity, diabetes, cardiovascular diseases and some types of cancer [10].

As recently reported, increased PA is favorably and strongly correlated with eating behaviors that are determined by oneself [11]. Increased PA is a key factor in facilitating the development of more positive eating behaviors and self-determined eating habits, which may help to reduce the risk of issues like obesity and eating disorders. On the other hand, however, regular PA seems to positively influence dietary behaviors by improving the sensitivity of the physiological satiety signaling system, modulating the requirements of macronutrients and food choices and modifying the hedonic response to food stimuli [12,13]. Furthermore, as argued by Drenowatz et al. [14], it appears that specific types of exercise influence the frequency and intensity of food cravings. A systematic review by Noll and colleagues highlighted that, although sports modalities have a major impact on nutrient intake, athletes do not modify their diet to meet the demands of the training load [15]. The authors, however, underlined that studies on athletes have focused more on nutrient consumption than on eating habits.

Given the impact that lifestyle can have on people’s health, the evaluation of the relationship between PA, SBs and eating habits may represent a research area that needs to be further explored.

Therefore, assuming that active and non-active individuals can adopt different health-related behaviors besides exercise, the aim of this study was to analyze SBs and dietary habits adopted by individuals regularly practicing exercise in a gym, sports athletes and inactive individuals. Possible differences among gender, age and type of PA or sport subgroups were also assessed.

## 2. Materials and Methods

During the period from January to May 2023, an anonymous online survey was used to collect sociodemographic and behavioral information among adult individuals from the Campania region, in the south of Italy. The survey was performed using the Google Module platform.

To enroll participants, QR codes were strategically placed in specific locations, including university classrooms, gyms, sports clubs, stores and restaurants. Each QR code was accompanied by a concise survey description. Participants accessed the questionnaire upon granting their informed consent to data treatment.

The Ethics Committee of the University of Campania “Luigi Vanvitelli” approved the protocol of this study (approval no. 28480/2022).

Age and gender were collected for each participant. Then, the PA level was investigated by using three questions: 1. “Have you been practicing physical activity or sports regularly for more than six months?”; 2. “If yes for how long?”; 3. “What kind of activities do you practice?” [16].

The Sedentary Behavior Questionnaire (SBQ) was used to evaluate the time spent in SBs, and the Healthy Dietary Habits Index (HDHI) was used to collect information about food consumption [17,18].

The SBQ measures the amount of time spent engaging in nine sedentary activities on an average weekday and weekend (watching television; playing computer or video games; sitting while listening to music; talking on the phone; doing paperwork or office work; reading; playing an instrument; doing arts and crafts; and driving/riding in a car, bus or train). We have made changes to the original version of the questionnaire by dividing the question aimed at investigating “the use of the PC in general and for playing video games” into two questions, one investigating video gaming and the other exploring the use of PCs to watch films, TV series and videos. The question “talking on the phone” was also changed specifically asking how much time participants spend on smartphones making phone calls or chatting, using social networks and watching videos. Finally, the question regarding arts and crafts was removed because this activity was included in the question regarding the sitting time to study, work or practice artistic activities. The SBQ asks participants how much time they spend “from when you wake up until you go to bed” engaging in each of the indicated SBs during a typical weekday and a typical weekend day. The response options were none, 15 min or less, 30 min, 1 h, 2 h, 3 h, 4 h, 5 h and 6 h or more. Each response was converted into hours, and the total numbers of hours of SB were summed separately for weekday and weekend days for each item. Next, weekly estimates were calculated by multiplying weekday hours by five and weekend day hours by two. Finally, another variable was created for the total number of hours spent in sitting behaviors per week. Answers representing more than 24 h/day were coded as 24 h/day [17].

The fifteen-item HDHI was used to investigate the frequency of consumption of red meat fat, chicken fat, low-fat foods, energy drinks, soft drinks, breakfast, fast foods, added salt, sweets, fruits and vegetables, fish, milk, fatty condiments and bread. Only one change was made to the question concerning fatty condiments, to which extra virgin olive oil was added. A score between 0 (least healthy) and 4 (most healthy) was allocated to each item, with a maximum reachable score of 60. For seven questions, the two healthiest answers were added together because they were considered equally healthy (trimming meat fat before consumption, trimming chicken fat before consumption, fish intake per week, use of low-fat products, fruit intake per day, vegetable intake per day, breakfast consumption per week) [18]. The higher the score was, the healthier the food habits.

### Statistical Analyses

This study followed a cross-sectional design. The software IBM SPSS version 28 for Windows (IBM Corp., Armonk, NY, USA) was used for the analysis. Statistical significance was considered when *p* < 0.05. The variables that had a normal distribution were expressed as mean and standard deviation. The ANOVA was performed to compare SBQ and HDHI scores among groups, using the post hoc Dunn–Bonferroni approach for pairwise comparisons. The chi-squared test was used to compare gender distribution and percentage of subjects choosing the healthiest habits to the HDHI questionnaire between groups. Furthermore, two multiple logistic regressions were performed to identify significant predictors of SB and dietary habits (SBQ total score, HDHI total score). Independent variables entered in the model were gender (0 = male, 1 = female), age (1 = lower than median value, 2 = higher than median value) and exercise practice (0 = inactive, 1 = gym practitioner/sports athlete).

## 3. Results

A total of 411 participants completed the questionnaire. The mean age of the total sample was 30.5 ± 13.3 years (18–73 years), 52% were females, 25% of participants were inactive, 34% were gym practitioners and 41% practiced different sports disciplines (gymnastics, athletics, cycling, dance, fencing, football, kickboxing, karate, running, skating, swimming and volleyball). Table 1 presents the main characteristics of the three groups. A higher proportion of males was observed in the athletes’ group, while the inactive group showed a higher mean age.

Table 2 shows the comparison of sedentary and dietary behaviors assessed through the two questionnaires among PA groups. As for the results of the SBQ, significant differences were found among groups considering the weekdays but not the weekend days. With regards to HDHI, significant differences were also observed. In particular, the post hoc analysis showed that these differences emerged from the comparison between inactive participants and athletes.

In detail, Table 3 shows the group differences related to the time spent in the different SBs investigated with the SBQ for weekdays. Watching TV, videos or series; using smartphones for chat and social media; reading; and sitting after moving from one place to another showed significant differences between groups. In particular, the inactive group spent more time watching TV with respect to the two groups of active individuals. Additionally, they used PCs to watch series and videos and read more than gym practitioners. In addition, inactive individuals made wider use of smartphones for chatting and social media and spent much more time to move from one place to another compared to sports athletes.

Considering the HDHI questions, significant differences were detected for type of bread, fruit, vegetable and salt consumption, as well as breakfast habits (Table 4). In particular, post hoc analyses revealed a significantly lower consumption of fast food, pizza and hamburgers (*p* = 0.048) and a higher choice of light products (*p* = 0.015) among gym practitioners compared to inactive participants, while athletes showed significantly higher consumption of fruits and vegetables compared to both inactive participants and gym practitioners (*p* = 0.011 and *p* = 0.002, respectively), and they reported consuming breakfast more than gym practitioners (*p* = 0.045). 

As for the regression analyses, age and exercising in a gym/sport were found to be inversely related to total sedentary time (Table 5) in fully adjusted regression models, while engagement in exercise in a gym or practicing sports was positively associated with the HDHI score (Table 6).

## 4. Discussion

Concerning the intricate connection between PA, SB, social media usage and dietary choices, the existing literature provides limited evidence, often concentrating on individual aspects of these relationships. In this study, health-related behaviors were compared between individuals practicing sports, attending gyms and not practicing any form of exercise. We aimed to explore the impact of a sedentary lifestyle on dietary choices but also to differentiate between various types of SBs. Moreover, we utilized a sedentary lifestyle questionnaire, which aided in identifying primary sedentary activities and gauging the extent of social media usage.

Our findings highlight several differences among these groups. First of all, inactive participants reported significantly higher amounts of weekly time spent in sedentary activities and significantly lower adoption of healthy dietary habits than active ones. Inactive individuals were more highly engaged in smartphone use and passive transport than athletes, in watching videos on a PC and reading than gym practitioners and in watching TV than both the other groups.

Notwithstanding these differences, it should be noted that for weekend days, total sedentary time, and for some specific habits, such as playing a musical instrument, studying or working, no differences were detected between the three groups. In line with previous studies, these outcomes show that sports athletes such as gym practitioners can be highly active and at the same time highly sedentary because of the independent relationship existing between exercise time and sitting time [19,20,21]. Indeed, people can be physically active and even achieve the recommended amount of PA but still be sedentary. This kind of SB can be considered a risk factor as well as inactivity. Therefore, in order to reduce its negative effects, it is even more important to comply with the PA recommendations [22]. Although attention towards the adoption of healthy lifestyles is growing over the last century, rapid technological and social changes are increasingly leading us to adopt behaviors characterized by low energy expenditure.

While various studies, both cross-sectional and longitudinal, suggest that SB is generally linked to negative physical and psychological consequences, it is important to consider that complete avoidance of SB in modern daily life may not be feasible [23,24]. Today, in fact, people frequently use SB to perform a variety of daily tasks, including working, driving, studying, eating and watching TV. Varied SB domains may have varied effects on mental health [25]. For example, screen-based SBs, which includes activities like watching TV, using a computer and using a smartphone or tablet, has been linked to detrimental impacts on mental health since it promotes social isolation and a sense of separation. A reduction in social support and communication may result from increased screen-based SB. Furthermore, certain screen-based behaviors might be considered largely passive activities that encourage indifference and favor other concurrent risky behaviors. Instead, by meeting fundamental social needs and offering opportunities for social support, certain screen-based SB may improve mental health [26].

Concurrently, SBs involving low-energy activities, primarily in a sitting or supine position, have been linked to the consumption of unhealthy foods in youth [27] but also in adults [3], as well as to an increase in obesity and a higher risk of chronic disease [28]. The most used metric for SB in earlier research has been the amount of time spent watching television, revealing links between TV viewing frequency and cardiovascular risk factors, obesity, metabolic syndrome and type 2 diabetes [29]. It has been demonstrated that watching TV increases calorie intake, also because of ads promoting foods high in fat, sugar and salt that are increasingly common when viewing TV, which encourage the consumption of these foods [30]. Furthermore, not only watching TV but also social media use may have some impact on teenagers’ and children’s dietary choices and desires [31,32]. In addition, the latest research in the field shows that healthy young adults may have unfavorable body images and eating choices as a result of social media activity or exposure to image-related content [33].

As for the dietary habits assessed in this study, inactive participants showed lesser attention to the choice of bread, lower intake of fruit and vegetables and less common consumption of breakfast during the week. Athletes, instead, were those who added salt less frequently to their plates. The association between healthier dietary habits and sports is not new in the scientific literature. In previous studies, athletes showed a nutritional status closer to dietary recommendations [34], better dietary habits [35] than sedentary subjects and even good adherence to the Mediterranean diet model [36,37]. Practicing a sport may allow athletes to make healthy dietary choices and reach the recommended nutritional status, balancing energy expenditure and intake.

In the regression analyses, exercise was negatively related to sedentary time and positively related to healthy dietary habits. These results confirm the differences found between active and inactive participants. However, another outcome could be worthy of concern. Although not significant, some differences were observed in both sedentary and dietary habits even between athletes and gym practitioners. This suggests that healthier habits could be associated with sports engagement more than with exercising in a gym, as previously reported [38].

However, it should be considered that this study did not take into consideration some variables that could be associated with the habits investigated. In particular, we did not explore the socioeconomic status of participants, which may be a determinant for engagement in sports, attendance of gyms or SBs. In fact, evidence shows that a sedentary lifestyle is associated with a low income [39,40]. Therefore, it is possible that our findings were in part related to the socioeconomic status of participants.

Additionally, concerns about the cost of healthy food can lead to differences in food purchasing based on household income [41]. Commonly, unhealthy behavioral clusters comprising low levels of moderate to vigorous PA, low consumption of fruits and vegetables and high screen time are prevalent in individuals from lower socioeconomic status [42].

Moreover, nutritional choices in the adult population are also dependent on educational attainment [43]. Therefore, even this variable could have influenced our results and should have been added to the regression models.

However, in light of these considerations, the lack of significant differences observed in the comparison of SBs assumed during weekend days by active and inactive individuals from our sample becomes more important. This could suggest that active behaviors are not adopted in daily life by athletes and gym practitioners beyond the time spent exercising. The use of accelerometers to assess the weekly PA level of individuals in future studies could help to better understand this finding [44].

Other than these limitations, some other aspects hinder the validity of this study. First of all, the width of the sample and the convenience sampling did not allow us to obtain a representative sample of the population examined. Furthermore, an electronic self-administered questionnaire was used to collect information. Therefore, it is possible that only individuals familiarized with the use of electronic media were involved, which limits the representativeness of the sample. Moreover, data were not objectively assessed, and a certain level of inaccuracy should be considered. Further research in this field, perhaps including analyzing more representative and wider samples through objective methods such as accelerometry, is needed to confirm our results. However, the sedentary and dietary behaviors of the three subgroups were examined in detail in our study, uncovering specific patterns associated with each group. Moreover, the findings of this study underline the need for enhancing the awareness of the local population regarding the detrimental role of sedentary time and unhealthy dietary behaviors, especially among inactive individuals.

## 5. Conclusions

This study delved into the comparison of health-related behaviors among individuals engaging in sports, attending gyms and those with no exercise routine. The findings revealed notable differences between these groups, particularly emphasizing the higher sedentary activity and poorer adoption of healthy dietary habits among inactive participants.

However, it is noteworthy that no significant differences were observed among the three groups for total sedentary time on weekends and certain specific habits, indicating the complexity of the relationship between PA and SB. This study highlighted the independent nature of exercise time and sitting time, illustrating that individuals can be physically active yet still engage in SBs, which poses its own set of health risks.

Furthermore, the dual nature of certain sedentary behaviors, which can either contribute to social isolation or fulfill basic social needs, was recognized, depending on the nature of the activity. Moreover, this study underscored the intricate relationship between health, PA, sedentary lifestyle and dietary habits. Social media and screen time were also implicated in shaping dietary preferences, especially among younger populations.

Although future research is needed to identify the determinants of these behaviors in the examined population, this study suggests that fostering awareness of health risks related to lifestyle, especially but not exclusively among inactive individuals, is needed for mitigating the detrimental effects of sedentary habits and unhealthy dietary choices.

## Figures and Tables

**Table 1 behavsci-14-00208-t001:** Demographic information and information related to exercise practice in the three groups of participants.

	Athletes N = 169	Gym Practitioners N = 140	Inactive N = 102	*p* Value
** *Gender M/F* **	98/71	63/77	36/66	0.001
** *Age* **	29.6 ± 13.6	28.1 ± 11.5	35.5 ± 13.8 *	0.001

* Inactive vs. gym practitioners and athletes.

**Table 2 behavsci-14-00208-t002:** Differences in SBQ and HDHI scores between the three participant groups.

	Athletes	Gym Practitioners	Inactive	*p* Value
** *SBQ weekdays (hours/day)* **	12.7 ± 6.8	13.9 ± 6.6	15.9 ± 8.6 *	0.001
** *SBQ weekend days (hours/day)* **	11.7 ± 7.1	12.2 ± 6.4	13.6 ± 8.5	0.115
** *HDHI* **	41.9 ± 4.5	40.7 ± 5.7	39.3 ± 6.6 *	0.001

* Inactive vs. athletes.

**Table 3 behavsci-14-00208-t003:** Mean values at the SBQ questions for each group expressed in minutes per day on a typical weekday.

Sedentary Behaviors (Minutes/Day)	Sport Athletes	Gym Practitioners	Inactive	*p* Value
** *Watching TV* **	49.6 ± 50.4	64.9 ± 65.9	97.3 ± 96.7 ^c^	<0.001
** *PC use to watch videos or series* **	52.6 ± 57.7	47.1 ± 52.6	68.3 ± 75.9 ^b^	0.033
** *Video gaming* **	25.1 ± 49.0	27.8 ± 60.9	24.8 ± 60.7	0.888
** *Listening to music* **	48.5 ± 77.0	36.4 ± 53.3	58.3 ± 96.2	0.083
** *Smartphone use to chat* **	116.1 ± 107.4	145.7 ± 118.6	155.6 ± 124.9 ^a^	0.012
** *Smartphone use to navigate social media* **	110.5 ± 105.7	136.3 ± 105.1	150.6 ± 122.1 ^a^	0.010
** *Smartphone use to watch videos* **	62.4 ± 81.9	71.6 ± 89.1	88.2 ± 104.1	0.077
** *Reading* **	49.7 ± 62.1	34.6 ± 45.1	55.3 ± 68.1 ^b^	0.014
** *Playing a musical instrument* **	8.2 ± 35.1	5.9 ± 35.1	2.8 ± 14.1	0.389
** *Sitting to move from one place to another* **	51.4 ± 53.6	63.5 ± 71.9	72.8 ± 93.8 ^a^	0.051
** *Sitting to study or for job* **	186.7 ± 124.7	198.3 ± 113.4	185.7 ± 141.5	0.661

^a^ Inactive vs. athletes; ^b^ inactive vs. gym practitioners; ^c^ inactive vs. athletes and gym practitioners.

**Table 4 behavsci-14-00208-t004:** Percentage of subjects who reported the healthiest habits to the HDHI questionnaire.

	Sport Athletes	Gym Practitioners	Inactive	*p* Value
** *Trimming red meat fat before consumption* **	63	61	60	0.350
** *Trimming chicken fat before consumption* **	75	63	69	0.888
** *Fish intake per week* **	36	32	30	0.721
** *Types of bread consumed* **	33	24	18	0.046
** *Soft drink consumption per week* **	51	64	52	0.076
** *Energy drink consumption per week* **	73	69	80	0.290
** *Sweets, snack and chocolate consumption per week* **	37	30	37	0.079
** *Purchasing fast food or takeaways such as hamburgers, fries, pizza* **	33	44	34	0.079
** *Types of fat spreads used* **	91	90	89	0.359
** *Use of low-fat products* **	53	59	44	0.335
** *Types of milk consumed* **	61	59	61	0.822
** *Fruit intake per day* **	34	32	25	0.016
** *Vegetable intake per day* **	48	40	34	0.006
** *Breakfast consumption per week* **	89	80	80	0.032
** *No salt added* **	31	12	16	0.001

**Table 5 behavsci-14-00208-t005:** Results of the logistic regression analysis with the total sedentary time as a dependent variable.

Explanatory Variables	Odd Ratio	95% Confidence Interval		*p* Value
		Lower Limit	Upper Limit	
** *Age* **	−0.931	0.394	0.258	<0.001
** *Gender* **	0.153	0.778	1.747	0.458
** *Exercise in a gym/sport* **	−0.884	0.252	0.678	<0.001

**Table 6 behavsci-14-00208-t006:** Results of the logistic regression analysis with the Healthy Dietary Habits Index as a dependent variable.

Explanatory Variables	Odd Ratio	95% Confidence Interval		*p* Value
		Lower Limit	Upper Limit	
** *Age* **	0.135	0.764	1.714	0.512
** *Gender* **	0.324	0.931	2.054	0.108
** *Exercise in a gym/sport* **	0.500	1.024	2.653	0.040

## Data Availability

The data presented in this study are available upon request from the corresponding author.

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
