# Peer review of "Sedentary Behaviors and Eating Habits in Active and Inactive Individuals: A Cross-Sectional Study in a Population from Southern Italy"

_behavsci, 2024, doi:10.3390/bs14030208_

Round 1
Reviewer 1 Report
Comments and Suggestions for Authors
Dear authors, thank you for this interesting study. However, with all due respect for the work carried out, we believe that it can and should be improved.
Namely:
2. Materials and Methods
It is not clear how the sample was chosen. Who did they send the questionnaire to? How many questionnaires did they send? What year and time of year was it held?
4.Discussion
The discussion should be more robust.
They state that in order to contain the length of the questionnaire, other socio-demographic and behavioral characteristics of participants, which could be related to the studied habits, were not explored. It is a justification that is difficult to accept, even more so when they are fully aware that, for example, the level of education, profession and socioeconomic level could be relevant to better understand and explain the results obtained.
Therefore, I consider that if it is not possible to use this type of variables, for the discussion to be more robust, it is not enough to make the reference they made to the limitations of the study, but rather, they should consider in more depth (for example using studies already carried out) in the discussion itself, the possibility that some specific results could be explained by some of the variables that were not taken into consideration.
Author Response
Dear authors, thank you for this interesting study. However, with all due respect for the work carried out, we believe that it can and should be improved.
We thank the Reviewer for the kind judgment. We acknowledge that the paper has important limitations. Thanks to the Reviewers’ indications, we have tried to address some of them.
Namely:
- Materials and Methods
It is not clear how the sample was chosen. Who did they send the questionnaire to? How many questionnaires did they send? What year and time of year was it held?
Thank you for your valuable feedback and for highlighting the omission in the Methods section. We appreciate your thorough review.
We have added the study period and the following sentence to the methods:
“To enroll participants, QR codes were strategically placed in specific locations, including university classrooms, gyms, sports clubs, stores and restaurants. Each QR code was accompanied by a concise survey description. Participants accessed the questionnaire upon granting their informed consent to data treatment.”
4.Discussion
The discussion should be more robust.
They state that in order to contain the length of the questionnaire, other socio-demographic and behavioral characteristics of participants, which could be related to the studied habits, were not explored. It is a justification that is difficult to accept, even more so when they are fully aware that, for example, the level of education, profession and socioeconomic level could be relevant to better understand and explain the results obtained.
Therefore, I consider that if it is not possible to use this type of variables, for the discussion to be more robust, it is not enough to make the reference they made to the limitations of the study, but rather, they should consider in more depth (for example using studies already carried out) in the discussion itself, the possibility that some specific results could be explained by some of the variables that were not taken into consideration.
Thank you very much for your insightful suggestions regarding the Discussion section, which indeed lacked thoroughness. We appreciate your recommendation to explore additional studies to enhance the discussion and analyze correlations with socioeconomic factors.
We understand the concern raised about the decision not to explore other socio-demographic and behavioral characteristics due to the questionnaire's length. Indeed, variables such as education, profession, and socioeconomic status, could contribute to a better understanding of the observed habits.
In response to your feedback, we have expanded the discussion by delving into relevant studies and considering the potential impact of unexplored variables on specific results.
Thank you once again for your thoughtful comments and constructive feedback.
Reviewer 2 Report
Comments and Suggestions for Authors
Good points are made about the distinction between sedentary behaviors and physical activity, which it the true highlight of the work.
All in all, I think the manuscript is well written.
The greatest concern I have is also noted on line 281-282. “Finally, in order to contain the length of the questionnaire, other socio-demographic and behavioral features of participants.” One of the biggest missing elements in the regression models is socioeconomic status. Participating in sport or being a member of gym has a cost, meaning that those in the inactive group likely had the lowest socioeconomic status. Socioeconomic status is related to dietary intake and may also impact breakfast consumption. So, it is unclear whether the associations in Table 6 are the result of physical activity or due to differences in socioeconomic status. Moreover, the p-value for physical activty is only 0.040, close to insignificance. Possible differences in socioeconomic status between the three groups need to be commented on in the paper. I suggest adding a paragraph to discuss the possible effect of socioeconomic status.
Table 1 – Why does the footnote read “*Inactive vs Gym practitioners and athletes.”? Should it not say significantly different than gym practitioners and athletes? Also, we know that the inactive group is inactive by its column title.
Table 2 – Why does the footnote read “Inactive vs athletes” when discussing the HDHI? Should it say Compared to athletes?
Table 3 – It is confusing to use #, *, and **. Use three different symbols or use both #* to note that the inactive group is different from the others.
Author Response
Good points are made about the distinction between sedentary behaviors and physical activity, which it the true highlight of the work.
All in all, I think the manuscript is well written.
We thank the Reviewer for the kind judgment. We acknowledge that the paper has important limitations. Thanks to the Reviewers’ indications, we have tried to address some of them.
The greatest concern I have is also noted on line 281-282. “Finally, in order to contain the length of the questionnaire, other socio-demographic and behavioral features of participants.” One of the biggest missing elements in the regression models is socioeconomic status. Participating in sport or being a member of gym has a cost, meaning that those in the inactive group likely had the lowest socioeconomic status. Socioeconomic status is related to dietary intake and may also impact breakfast consumption. So, it is unclear whether the associations in Table 6 are the result of physical activity or due to differences in socioeconomic status. Moreover, the p-value for physical activty is only 0.040, close to insignificance. Possible differences in socioeconomic status between the three groups need to be commented on in the paper. I suggest adding a paragraph to discuss the possible effect of socioeconomic status.
Thank you very much for your thoughtful and constructive feedback on the Discussion section. We truly appreciate your insightful comments, particularly concerning the need for a more comprehensive exploration of socioeconomic factors.
We agree that addressing this aspect is crucial for a more nuanced interpretation of the results.
In response to your recommendation, we have included a dedicated paragraph in the discussion. Your guidance is instrumental in refining the manuscript, and we are grateful for your thorough review.
Thank you once again for your time and valuable insights.
Table 1 – Why does the footnote read “*Inactive vs Gym practitioners and athletes.”? Should it not say significantly different than gym practitioners and athletes? Also, we know that the inactive group is inactive by its column title.
Table 2 – Why does the footnote read “Inactive vs athletes” when discussing the HDHI? Should it say Compared to athletes?
Table 3 – It is confusing to use #, *, and **. Use three different symbols or use both #* to note that the inactive group is different from the others.
With the abbreviation “vs” we meant to indicate that the observed differences among the three categories were due, as showed by the post-hoc analysis, by those specific comparisons: inactive versus athletes, inactive versus gym practitioners, or inactive vs athletes and gym practitioners.
In order to increase the clarity of these results, we have tried to better specify these concepts in the text and we have changed the symbols in Table 3.
Round 2
Reviewer 1 Report
Comments and Suggestions for Authors
I consider that the changes made by the authors have respected what was suggested and have improved the article. I therefore consider that it fulfils the conditions to be published.
Reviewer 2 Report
Comments and Suggestions for Authors
The additions made to the end of the work, give it more subtly, complexity, and depth to work.